# Natural Additives Improving Polyurethane Antimicrobial Activity

**DOI:** 10.3390/polym14132533

**Published:** 2022-06-21

**Authors:** Natalia Sienkiewicz, Sylwia Członka

**Affiliations:** Institute of Polymer & Dye Technology, Faculty of Chemistry, Lodz University of Technology, 90-537 Lodz, Poland; sylwia.czlonka@dokt.p.lodz.pl

**Keywords:** polyurethane foams, antibacterial properties, natural additives

## Abstract

In recent years, there has been a growing interest in using polymers with antibacterial and antifungal properties; therefore, the present review is focused on the effect of natural compounds on the antibacterial and antifungal properties of polyurethane (PUR). This topic is important because materials and objects made with this polymer can be used as antibacterial and antifungal ones in places where hygiene and sterile conditions are particularly required (e.g., in healthcare, construction industries, cosmetology, pharmacology, or food industries) and thus can become another possibility in comparison to commonly used disinfectants, which mostly show high toxicity to the environment and the human health. The review presents the possibilities of using natural extracts as antibacterial, antifungal, and antiviral additives, which, in contrast to the currently used antibiotics, have a much wider effect. Antibiotics fight bacterial infections by killing bacteria (bactericidal effect) or slowing and stopping their growth (bacteriostatic effect) and effect on different kinds of fungi, but they do not fight viruses; therefore, compounds of natural origin can find wide use as biocidal substances. Fungi grow in almost any environment, and they reproduce easily in dirt and wet spaces; thus, the development of antifungal PUR foams is focused on avoiding fungal infections and inhibiting growth. Polymers are susceptible to microorganism adhesion and, consequently, are treated and modified to inhibit fungal and bacterial growth. The ability of micro-organisms to grow on polyurethanes can cause human health problems during the use and storage of polymers, making it necessary to use additives that eliminate bacteria, viruses, and fungi.

## 1. Introduction

Nowadays, disinfectants are mostly used for their cost-effectiveness and powerful outcomes, but these chemicals show high toxicity to the environment and human health. Such measures have many disadvantages, e.g., skin irritation on prolonged contact, possible bronchial irritation from inhalation, general caustic effects and corrosion of metals, product deterioration on standing, and others cause those other solutions are currently being sought. Natural antimicrobial additives can be successfully an alternative to chemical antimicrobials. Different kinds of plants, herbs, and spice extracts that have been used in natural medicine for many years have the greatest antimicrobial activity. The bio-resistance potential of clove, oregano, curcumin, cinnamon, birch juice, and thyme essential oils and components, cinnamaldehyde, eugenol, carvacrol, and thymol have been confirmed in the literature [1,2]. Many studies show that natural extracts with antibacterial properties can be successfully used as additives to polymers influencing their final properties. The most important feature of materials with antibacterial properties should be to inhibit the appearance, and growth of bacteria and preclude their cumulation. The largest amounts of the most popular bacteria, such as *Escherichia coli*, *Klebsiella pneumoniae*, and *Staphylococcus aureus*, appear on objects which are near to patients, such as bed rails and headboard, armchairs, and cabinets so there are other contact surfaces, from which bacteria can easily transfer to patients and their guardians and other employees [3,4]. In the presented works, there are used as additives and modifiers of polyurethanes (PUR), which there are thoroughly researched and used as natural medicines daily. There are no reports in the literature of the use of other toxic natural extracts as polyurethane modifiers. Before starting the study, data on the toxicity of the selected plant should be collected. If the plant is not toxic, continue testing if no toxicity data exist; appropriate determinations should be selected for the toxicity analysis. It is also important to develop and prepare a safety and toxicity protocol. Additionally, it should also be mentioned that the type of polymers used has significant importance in the search for final products that should mainly inhibit the growth of bacteria, fungi, and viruses. It is effectual that the polymers selected to produce specific items should have a bacteriostatic activity to prevent the expansion of bacterial colonies in their proximity or work bactericidal and radiogenic impair nearby bacteria, by changing the course of their cell processes. Important characteristics to keep in mind are the yield and the type of surface to be used. The area with antibacterial features should be prepared with polymers which properties have a long useful time with contemporary high productivity of effect. Moreover, these polymers should show refractory to outside agents so that the process of leaching the antiseptic does not appear because of chemical or mechanical effects and it should also have the odds of constant bonding to different surfaces [5,6,7,8].

PUR foams are mainly used as insulation materials. Therefore, they should be resistant to viruses, bacteria and fungi. Commercial, synthetic compounds used as antimicrobial and antioxidant agents are mainly derived from phenol, and they are heavily toxic for human health. Nowadays, the tendency is to limit the use of synthetic additives, and replace them with the natural, bio-based antimicrobial and antioxidants additives that do not impact the human health. The application of natural compounds with an antimicrobial and anti-aging activity have been used in previous research. For example, Członka et al. [9] studied the antioxidant and antimicrobial activity of PUR foams containing clove filler. The obtained materials showed great antibacterial activity against selected bacteria, e.g., *E. coli* and *S. aureus*. Similar results were obtained in the case of PUR foams with incorporated nutmeg filler—the inhibitory zone of four bacterial strains (*E. coli*, *S. aurous*) significantly increased with the incorporation of nutmeg compound [10]. The antimicrobial activity of cinnamon extract was also confirmed by Liszkowska et al. [11] in the case of new materials based on cinnamon extract embedded in PUR matrices. The presented results confirmed that the natural, antibacterial compounds can be incorporated into PUR foams and retain their inhibitory effect against microbial growth. Previous studies have shown that strong interfacial interaction, such as hydrogen bonding, can be easily formed between the additive molecules and isocyanate groups, leading to the formation of a cross-linked structure. It has been shown that the functional groups of natural additives (e.g., hydroxyl and amine groups) can react with isocyanates even in the absence of catalyst leading to the formation of urea bonds [9,10,11,12,13]. The impact of the addition of natural additives on the cross-linking density of polyurethane composites is presented in Figure 1.

Due to the constant interest in PUR materials with antibacterial properties, the prepared work contains the latest information on this topic and, in order to be readable for the recipient, has been divided into two main parts, such as antimicrobial PUR and antimicrobial PUR foams. The first chapter describes all forms of polyurethanes, except for foams, which, due to their wide application in many fields, describes the next chapter. The presented review shows that incorporation of natural modifiers with given antimicrobial properties into conventional PUR and PUR foam formulations obtain materials with assumed features.

## 2. Polyurethanes Modified with Antimicrobial Compounds

Recently, PUR is widely used in the production of equipment that outright comes in contact with a person as medical application and others [15,16,17,18]. Results in the literature indicate that the possibility and capacity of bacteria, viruses or fungi increasing on the surface of PUR can initiate a human health problem that arises during the use and storage of these materials.

A good example to raise antiseptic properties of PUR is to change them with the application of substances, which expose antimicrobial features like cinnamaldehyde (CA). Kucinska-Lipka and Feldman shown the synthesis of antibacterial porous structure of PUR matrices (MPTLs) in a shape of fine layers by applying solvent-casting/particulate-leaching method jointed with thermally initiated phase separation [19]. Furthermore, received MPTLs were modified with cinnamaldehyde at various percentage concentrations (0.5–5%) to create the best antimicrobial result of the CA applied into the MPTLs structure. Both unmodified and CA-modified MPTLs samples were defined by mechanical and physicochemical properties as well as by assessment of their antibacterial efficiency. The obtained results shown that the most important antibacterial effect of CA-modified MPTLs was received when the CA amount was 3.5% and it was between all concentrations used. All the described properties arise from the fact that the CA exhibits antibacterial, antifungal, and anti-inflammatory properties. CA in other words, cynamal, is the yellow liquid with an intensive and sweet-spicy flavor, with the group of aromatic aldehydes [20]. The greatest content of CA is the oil of *Cinnamomum zeylanicum* tree [21]. Because of that CA has in its structure aldehyde groups show that it can be built in the chemical structure of such materials as polyurethane, chitosan, polylactide, cellulose, or alginates. The meaningful advantage of speaking in favor of this substance is the fact that it has been accepted as commonly safe and harmless by the Food and Drug Administration in the USA [22]. Scientists’ reports confirm and clearly indicates preservative properties of CA towards antimicrobial activity against *Pseudomonas syringae* pv. *actinidiae* (plant pathogen) [23], *E.*
*coli*, and also *Salmonella* [24]. Importantly, with regard to the quoted studies, CA after inclusion into cellulose-based package film also had antimicrobial activity against pathogenic bacteria such as: *Aeromonas hydrophila*, *Bacillus cereus*, *E. coli* DMST 4212, *E. coli* O157: H7, *Listeria monocytogenes*, *Micrococcus luteus*, *Pseudomonas aeruginosa*, *Salmonella enteritidis*, *S. aureus*, *Enterococcus faecalis*, and antifungal working against three yeasts *Candida albicans*, *Saccharomyces cerevisiae*, and *Zygosaccharomyces rouxii* [25]. The results indicate that the mechanism of antibacterial and antifungal action of CA depends on its concentration in material. Generally, trans-cinnamaldehyde can impede the increase in *E. coli* and *S. typhimurium* without breaking up the external coating of a bacterial cell or emptying intracellular Adenosine Triphosphate (ATP) Concentration. Research shows at low concentrations, it restrains enzymes committed in cytokine mutuality or different significant cell functions but when the concentration is higher, it restrains ATPase, and at deadly quantities, cinnamaldehyde set whirling the construction of cell membrane [26]. The mechanical features of samples and morphology advisable that CA uncovers an effect on these properties, but the contact angle of the area has little increased for the reason of CA hydrophobic character. Noticed mass change (over 50% for unmodified and almost over 60% for CA-modified MPTL) shows that these proposed material compositions may find a successful use as biodegradable coating. In accordance with the executed research, it is supposed that received CA-modified MPTLs can find a common use as functional antibacterial matrices for reclamation of injured and damaged skin. Purposeful tests as Tensile strength (TSb), elongation at break (εb), and Young’s modulus (E) were made and the results are as follows. Unmodified MPTLs had TSb of 0.32  ±  0.01 MPa, εb of 78  ±  6%, and E of 0.254  ±  0.15 MPa but which is very interesting modification with CA only slightly affected on the mechanical properties of CA-modified MPTLs. The TSb, εb, and E of CA-modified MPTLs decreased to 0.18  ±  0.02 MPa, 68  ±  4%, 0.244  ±  0.09 MPa, accordingly [27].

The antiseptic and simultaneously anti-adhesive properties of PUR have been obtained by immobilizing chitosan and heparin on the samples of PUR specially prepared for this purpose via a stepwise method. Antibacterial functions of materials were obtained by plasma and modified with different concentrations of chitosan (0.5 and 2.0%) and heparin immobilization. Prepared samples for analyses were modified to be antibacterial resistant against *S. aureus*, *S. epidermidis* (both Gram-positive), *E. coli*, and *P. aeruginosa* (both Gram-negative) bacteria. The obtained bacterial adhesion results were satisfactory and indicated a meaningful decrease in the number of feasible bacteria on alike samples modified chitosan as well as modified heparin where for these samples the results were better and the most efficient (Figure 2 allows better perceive the presented works and there are not as accurate data and results) [28]. Kara et al. [29] showed that the polyurethane surfaces with towering hydrophilicity and surface free energy-adjusted anti-adhesion efficiency versus bacteria. Additionally, stability studies of the obtained systems are presented, which indicate that immobilizing chitosan as well as immobilizing heparin were stable and did not separate from the polyurethane surfaces. Moreover, the results showed that chitosan and heparin were covalently linked to the polyurethane by activating the surface with plasma and glutaraldehyde which ensured reactive aldehyde groups that can simply react with the amine and hydroxyl groups of chitosan and heparin [29].

New antibacterial wound dressing materials were prepared via coating solution with *Szygium* aromatic extract (clove oil) thermoplastic polyurethane nanofibers mats. Used clove oil was obtained by Soxhlet extraction and subjected to further examination by gas chromatography-mass spectrometry analysis, which confirmed that the extract was primarily assembled of eugenol and β-caryophylene with preferable antimicrobial activity. Polyurethane materials were coated with early received clove oil to obtain nanofibers with assumed antibacterial properties. Research proves that even 2 mg cm^−2^ clove-oil-coated polyurethane nanofibers show an area of security against *S. aureus* and *E. coli*, respectively. The inhibition area results are shown in Table 1 [30].

The work also noted that air permeability of nanofibrous materials reduced with the growing quantity of clove oil over 5 mg cm^−^^2^, which is due to those clove oil caused modifications in the morphology of nanofibers. It is worth pointing out, that the inherence of clove oil did not affect the morphology of nanofibers coated with 2 mg cm^−^^2^ clove oil and, in this case, the air permeability utility of those bandages was close to that of unmodified nanofiber materials [30].

Yue et al. [31] presented the possibility of using thymol loaded ethanol-soluble polyurethane as nanofibrous membranes with antibacterial activity. Thymol is a natural monoterpenoid phenol derivative of p-Cymene, isomeric with carvacrol, found in oil of thyme, and extracted from *Thymus vulgaris* (common thyme), *Ajwain* (common cumin), and various other kinds of plants as a white crystalline substance of a pleasant aromatic odor and strong antiseptic properties. Scientists obtained a dressing material with complex properties with antibacterial activity, waterproofness, and water vapor permeability used polyurethane-Thymol nanofibrous membranes with various amounts of thymol (2, 4, 6, and 8 wt.%) to determine the effects of thymol on the morphology and execution of fibrous membranes. Finally, the resultant nanofibrous membranes composed of PUR, fluorinated PUR, and thymol had homogeneous structure, good water resistance with the hydrostatic pressure of 17.6 cm H_2_O, great breathability of 3.56 kg m^−2^ d^−1^, the significant tensile stress of 1.83 MPa, and tensile strain of 453%, and very promising antibacterial activity. In the presented research, it was also important to search and define the antibacterial activity of polyurethane-Thymol membranes against *S. aureus* and *E. coli* based on the method of colony counting. The control plates for nonmodified PUR samples showed crowded colonies of bacteria, whereas the PUR modified with 8 wt.% Thymol plates did not show any colony. It is also worth noticing that all polyurethane-Thymol nanofibrous materials revealed antibacterial activity. Additionally, to investigate the germicidal effects of polyurethane-Thymol materials, the structural changes of *E. coli* and *S. aureus* were evaluated by SEM. For PUR modified by 14 wt.% Thymol, *E. coli* cells were lifeless and shown a specific rod-shaped structure. Nevertheless, for PUR modified by 8 wt.% Thymol, the cells of *E. coli* were spoiled, and the cytoplasm issued out what confirmed that the cell membranes were damaged. After the modification with 14 wt.% of Thymol, both *E.*
*coli* and *S. aureus* had spherical sleek cells. After the modification with 8 wt.% of Thymol, the morphology of the cells became more erratic. To confirm the obtained results, fluorescence-based live/dead bacteria designation was also applied. Calcein, used as a non-fluorescent cell penetrable dye, is enzymatically transformed to potent fluorescent calcein to recognize existing bacteria. Increased red fluorescence was observed in the case of PUR modified with 8 wt% of thymol. This confirmed the antimicrobial properties of the modified PUR fibrous material. Additionally, developed by researchers the in situ electrospinning by the designed mobile equipment showed the potential to develop convenient and preventive fibrous materials for the skin at any time, which will promote the common use of electrospun fibrous injury bandage materials. Scientists used in their research 3 wt% *Salvia* against *S. aureus*, *E. coli*, and *P. aeruginosa* [31].

Another example of the use of natural additives with antibacterial properties in polyurethane materials was presented by Santamaria-Echart et al. [32]. Their work was focused on the preparation of Salvia-based waterborne polyurethane-urea (WBPUU) dispersions with cellulose nanocrystals (CNC) allowing the preparation of functional green nanocomposite films with enhanced mechanical properties. The work showed that *Salvia officinalis* extracts, recognized for their antibacterial and antioxidant properties [33,34,35,36], were able to transfer the same properties to the nonmodified WBPUU. Furthermore, during the microorganism test, the effect was positive against Gram-positive *S. aureus*, and Gram-negative *E. coli,* and *P. aeruginosa* bacteria. It was noticed that after an incubation period of 1 day, Salvia-based WBPUU films showed a bacteriostatic effect against chosen bacteria, impeding their growth. What is more, after an incubation of 4 days the detention power of the nonmodified WBPUU material was discontinued for all the attempted bacteria, where the modified with natural extract WBPUU films showed individual behaviors dependent on the used incorporation of antibacterial extract. As concerns Gram-positive *S. aureus*, it was shown that the arresting effect caused by the adding of *Salvia* extract was only efficient in the case where the in situ method was used and applied amounts 3 and 5 wt.% of extract. For samples with *Salvia* extract, the bacteria growth was inhibited in most of the modified polyurethane films confirming their yield against this bacterium. The sample containing 1 wt.% natural extract showed slightly different results and the inhibition effect was impeded. A similar tendency was noticed against *P. aeruginosa*. Moreover, considering that *P. aeruginosais* is more resistant, if compared with *E. coli* (being Gram-negative bacteria), the inclusion through the post-method did not score effective, presumably on account of lixiviation effects further time [32].

Other obtained results showed the low and tight particle size distribution of the Salvia-based WBPUU dispersion, which can be associated with the effective synthesis process with the natural extract supplement. The dispersion was used to obtain novel organic nanocomposites by introducing isolated cellulose nanocrystals with large length-to-diameter ratio. The aspect ratio value is a really important parameter setting the efficiency of the CNC in the nanocomposite. The higher this ratio is, the greater the reinforcing capacity of the system. This worth is in the top scope comparing with congruous materials, take advantage of the reinforcing effect it can transfer to obtained Salvia-based WBPUU samples. What is more, FTIR analyses confirmed that the addition of CNC caused the increase in both, free C=O groups in the soft segment of urethane and hydrogen-bonded C=O groups of urea groups. Differential scanning calorimetry (DSC) results took notice a process of changing how phases induced by CNC incorporation is organized that, in general, resulted in lower the mobility of the soft domain values (indicating the increased mobility of the amorphous fraction of soft domains), while favoring the system of brief range ordered hard spheres, indicated by the increase in ΔH_mHS_ values. The relations between the parts or elements accepted by the WBPUU, also with the rigid CNC, make it possible to receive composites with stable mechanical properties (higher E), and extension elongation ability [32,37].

A new not harmful to living tissue and antimicrobial material with effective wound healing activity can be a specific option for wound dressing use. Eskandarinia et al. [38] presented polyurethane-hyaluronic acid (PU-HA) nanofibrous wound dressing with three different amounts of ethanolic extract of propolis (EEP). Propolis (bee glue) is a piceous beehive residue collected by honeybees from different plants and next there is applied to the gasket in honeycombs, polish the inside walls and also protect the entry against insects. A lot of pharmaceutical materials including antibacterial, antifungal, antiviral, antiprotozoal, anti-inflammatory, antioxidant, anti-cancer, and other activities have been prepared from propolis [39,40,41,42]. Ethanol extract of propolis indicates various antimicrobial facilities according to the type of the tested drain off and activity time. This extract of propolis proves severe antimicrobial activities towards Gram-positive cocci strains belonging to *S. aureus* types [35].

The new propolis modified PUR-HA nanofibrous were featured by attenuated total reflectance/Fourier transform infrared spectroscopy, thermal gravimetric analysis, scanning electron microscopy, mechanical investigations, antimicrobial tests, and others. The PUR-HA/1% EEP and PUR-HA/2% EEP ones indicated higher antibacterial activity against *S. aureus* (2.36 ± 0.33 and 5.63 ± 0.87 mm) and *E. coli* (1.94 ± 0.12 and 3.18 ± 0.63 mm) in comparison with other tested samples. On the other hand, the PUR-HA/1% EEP sample shown notably higher biocompatibility for L929 fibroblast cells in comparison with PUR-HA/2% EEP. Additionally, the PUR-HA/1% EEP sample could crucially speed up the wound healing advancement and wound closure at the animal type. At the histopathological analysis, revised skin growth and collagen deposition at the healed wound place of the PUR-HA/1% EEP sample in comparison with other groups was observed. The obtained results showed that 1 wt.% EEP enriched PUR-HA nanofibrous can be a sanguine material with substantial biocompatibility, wound healing, and antibacterial features for next biomedical uses [38,43,44].

In the study presented by Shababdous et al. [45] two types of PUR were prepared in a two-step polymerization based on polycaprolactone, hexamethylene diisocyanate, and 1,4-butandiol. Obtained results of FTIR analyses and ^1^H NMR spectroscopy characterization indicated the achievement of PUR synthesis. Furthermore, thermal features of the PUR were carried by DSC and the obtained effects indicated that soft segments of analyzed PUR are amorphous and solid units of this material are crystalline. Additionally, the hydrophilicity of the presented materials was studied, and the outcomes showed that curcumin-loaded polyurethane is more hydrophilic than nonmodified PUR. Curcumin as a modifier was applied in the polyurethanes by electrospinning process in various amounts. The lowering speed of curcumin from polyurethane samples as well as antimicrobial activity of the matrix was explored and the results have shown that curcumin-modified PUR had satisfying mechanical and anti-bacterial properties, which confirmed that this material are the right candidate for wound dressing uses [45].

To heal diabetic wounds, PUR/carboxymethylcellulose nanofibers containing *Malva sylvestris* extract are used. Malva extract as a modifier was incorporated in the PUR and different amounts of carboxymethyl cellulose blend and has been observed utility, antibacterial, and wound healing properties [46,47,48,49,50]. The obtained research results indicated that increasing the diameter of the nanofibers and decreasing the carboxymethyl cellulose concentration raised in progressive the freeing time up to 85 h. The application of natural extract in the polymer material extended the mechanical and thermal properties of nanofibers. FTIR analysis showed the generating of hydrogen bonds between the extract and used PUR. The cell culture investigation was conducted by the protocol reported by Almasian et al. [51] and applied analysis indicated the non-toxic nature of produced wound dressings. Moreover, the extract containing nanofibers (15% w/w) exhibited 70.66 and 69.83% antibacterial activity against *E. coli* and *S. aureus* bacteria, accordingly. The wound dressing comprising 15% natural Malva extract presented the wound healing rapidity of 95.05 ± 0.24% by day 14. Additionally, from the histological images, it was noted that macrophage infiltration, neovascularization activity, and fibroblastic proliferation increased on the 7th day and the grade of collagenization and epithelium regeneration was increased on the 14th day [51,52].

Table 2 summarizes the methods of polyurethane modification to obtain antibacterial properties of the material.

## 3. Polyurethane Foams Filled with Antimicrobial Compounds

Antimicrobial added ingredients have been also applied in the synthesis of rigid PUR foams. Liszkowska et al. modified rigid polyurethane-polyisocyanurate foams (RPU/PIR) with cinnamon extract, green coffee extract, and cocoa extract in an amount of 10 wt.% [53]. All used natural additives such as cinnamon extract, green coffee extract and pine oil have antimicrobial characteristic [54,55,56]. All prepared samples of foams were submitted to climatic degradation in the special chamber with determined temperature, humidity, and UV radiation for 7, 14, and 21 days. Next the physico-mechanical features of foams were tested. Research results indicated that the compressive strength of foams reduced with the extended time of foam thermal degradation. The analyzed foams after degradation were obfuscated and changed the color became more red, light brown and yellow. The obtained results have shown that the addition of coffee and cocoa extract increased the aging resistance of foams (∆m was less than 1%) in comparison with sample without a filler and foam with added cinnamon extract. The compressive strength (CS) for degraded foams (maximum degradation time 21 days) decreased what there was approved on SEM analysis that foams modified with plant-based fillers (spherical shape of the foam cells) and standard foam were degraded under the effect of UV radiation, moisture, and temperature in a similar way. It is worth noting that biodegradation tests have proved that foams modified by cinnamon, green coffee, and cocoa extracts were more liable to biodegradation than the reference foam. The samples were distinguishable by approximately seven-times higher biochemical oxygen demand (BOD) values after 28 days of biodegradation than the nonmodified foam, which have less than 10 mg/L lower theoretical oxygen demand (TOD) values and 7–9 times upper degree of biodegradation (Dt) values [53,54,55,56].

### 3.1. Polyurethane Foams Modified with Lavender Filler

In another study, 2 wt.% of lavender functionalized with kaolinite (K) and hydroxyapatite (HA) were added to PUR samples [13]. Thus, prepared lavender fillers were used as a reinforcing material in the synthesis of PUR foams. The reference foam was labeled as PUR_REF, whereas the PUR composites reinforced with non-functionalized lavender, lavender functionalized with kaolinite, and lavender functionalized with hydroxyapatite were labeled as PUR_L, PUR_L_K, and PUR_L_H, respectively.

It was noticed that the modification of lavender using a high-energy ball milling process influence the exterior morphology and size of filler particles. In Figure 3, the morphology of non-functionalized lavender filler was shown, and the sample contained lavender particles between 950 nm and 3 µm with a medium value at ~1.5 µm. The application of the functionalization with kaolinite and hydroxyapatite caused the structure of lavender particles to become more uniform and smoother, whereas the average size of filler decreased to 712 and 615 nm, properly [13].

Considering the antibacterial and antioxidative properties of lavender [57,58], kaolinite, and hydroxyapatite, the antibacterial properties of PUR composites filled with lavender fillers against *E. coli*, *S. aureus*, *B. subtilis*, *C. albicans*, and *A. niger* were evaluated (Table 3). The obtained results confirmed the antibacterial activity of PUR composites against bacteria, but no activity against fungi was observed. The authors reported that a low antibacterial activity against fungi may relate to low concentration of lavender fillers in the PUR composites.

Furthermore, the effect of fillers modified with lavender on mechanical properties was defined by examination the compressive strength (σ_10%_), flexural strength (σ_f_), and impact strength (σ_I_). The results clearly showed that the application of non-functionalized and functionalized lavender fillers influences the value of σ_10%_. When the reference sample is compared, σ_10%_ increased by ~7, ~15, and ~17%, for PUR composites reinforced with non-functionalized lavender, lavender functionalized with kaolinite, and lavender functionalized with hydroxyapatite, respectively. An analogy trend was observed in the case of σ_10%_ measured perpendicular to the direction of the foam expansion—the value of σ_10%_ increased by ~8, ~18, and ~16% for with non-functionalized lavender, lavender functionalized with kaolinite, and lavender functionalized with hydroxyapatite, respectively. Research author for the purpose of securing of seeming density on the mechanical characteristics of PUR samples specified the compressive strength as well. In accordance with the shown results, the mechanical strength of polyurethane composites little increased—the specific strength (measured parallel) measured for PUR_REF was 6.5 MPa kg^−1^ m^−3^, whereas due to the application of lavender fillers, the value increased to 6.8, 7.1 and 7.3% for with non-functionalized lavender, lavender functionalized with kaolinite, and lavender functionalized with hydroxyapatite, respectively (Figure 4). Furthermore, the amplification effect of lavender modified fillers was also proven by the σ_f_ and σ_I_ results. Analyzing the PUR_REF foam, the application of non-functionalized lavender filler causes growth the value of σ_f_ by ~5%, and the use of lavender filler functionalized with kaolinite and hydroxyapatite enhanced the value of σ_f_ by ~9 and ~12%. A very similar trend was visible for σ_I_. Better results were observed for PUR samples reinforced with lavender filler functionalized with kaolinite and hydroxyapatite—the value of σ_I_ increased by ~4 and ~7%, respectively.

The flame retardant properties of such developed PUR composites were performed using a cone calorimeter test (Table 4). The results of ignition time (IT), peak heat release rate (pHRR), total smoke release (TSR), total heat release (THR), the average yield of CO (COY), and CO_2_ (CO_2_Y), and limiting oxygen index (LOI).

According to the presented results, the addition of lavender fillers affected the value of peak heat release rate (pHRR). Among all modified series of PUR composites, the lowest value of the pHRR parameter was observed for the PUR composites reinforced with lavender functionalized with hydroxyapatite, which was over 50% lower than for the PUR_REF. Furthermore, the incorporation of each filler resulted in a lower value of total smoke release (TSR). When compared with PUR_REF, the value of TSR decreased by ~7% for PUR composites reinforced with non-functionalized lavender, ~29% for PUR composites reinforced with lavender functionalized with kaolinite, and ~30% for PUR composites reinforced with lavender functionalized with hydroxyapatite. All results pointed that the application of lavender fillers saves the polyurethane chemical structure from next combustion and precludes heat transfer through the matrix of PUR sample. The addition of lavender fillers reduced the value of total heat release (THR). For PUR_REF sample the value of THR was 21.5 MJ m^−2^, but the addition of lavender fillers reduced the value of THF parameter to 21.1 MJ m^−2^, 20.5 MJ m^−2^, and 19.8 MJ m^−2^ further PUR composites reinforced with non-functionalized lavender, lavender functionalized with kaolinite, and lavender functionalized with hydroxyapatite, respectively. Moreover, the inclusion of lavender fillers changed the carbon monoxide (CO) to carbon dioxide (CO_2_) ratio, which is related to foam toxicity. The application of lavender fillers successfully raises the limiting oxygen index (LOI). The best results were observed for PUR samples modified with lavender functionalized with hydroxyapatite and kaolinite—the LOI index growth from 20.2% to 22.7% and 22.2%, accordingly.

The char residue of PUR composites after the combustion process was evaluated using scanning electron microscopy. According to the images presented in Figure 5, after the combustion process, the char residue of the reference PUR foams was loose and possessed a few fragments that were formed during the decomposition process. PUR composites reinforced with lavender fillers presented more compact char residue. The authors explained that a denser structure of PUR composites may act as a physical barrier, effectively limiting the heat transfer through the PUR structure and successfully inhibiting the combustion process.

### 3.2. Polyurethane Foams Modified with Nutmeg Filler

In another study, PUR composite foams were successfully reinforced with different concentrations (1 wt.%, 2 wt.%, 5 wt.%) of nutmeg filler. The reference foam was labeled as PU_N_0, whereas the PUR composites reinforced with 1, 2, and 5 wt.% of nutmeg fillers were labeled as PU_N_1, PU_N_2, and PU_N_5, respectively. The impact of natural nutmeg as filler on mechanical, thermal, antimicrobial, and anti-aging features foam samples was examined [10].

The SEM figures of sample shown that the inclusion of nutmeg filler led to the formation of an inhomogeneous structure with a major number of open cells. In Figure 6 and Figure 7, it can be observed that the addition of 1 and 2 wt.% of nutmeg did not change the structure morphology of PUR foams. The closed-cell formation of PU_N_1 and PU_N_2 was correctly maintained, but the sample with addition of 5 wt.% of nutmeg had the less homogeneous structure, and the number of defective cells was significantly greater. In relation to the reference PU_N_0 foam, the porosity of polyurethane samples was lightly reduced for example for PU_N_5 composition, the porosity lowered from 87% to 79% [10].

Most importantly, it has been found that the addition of nutmeg filler effectively improved the antibacterial properties of such developed PUR composites. Table 5 shows obtained research results of the antibacterial activity of polyurethane samples against *S. aureus* and *E. coli*. The meaningful stopping of effect selected bacteria was shown for all PUR samples after 24 h of endangering. The composite with added 5 wt.% of nutmeg filler had the best antimicrobial properties. Good levels of antimicrobial activity against Gram-positive and Gram-negative bacteria were imputed to the different extracts and the essential oil contained in nutmeg seeds among others trimyristin, myristic acid, α-pinene, β-pinene, p-cymene, β-caryophyllene, and carvacrol [59,60].

The color analysis is shown in Table 6. The addition of nutmeg affected the color of the PUR samples. All examined PUR samples filled with nutmeg filler possessed a more intensive color, which was affirmed by the reduced value of lightness (L*). Furthermore, the analyzed PUR samples had a more intensive shadow of yellow (increased value of a*) and red (increased value of b*), which confirmed the presence of organic substances, such as polyphenols, quinones, or flavonoids of nutmeg seeds [59,60].

PUR composites were subjected to the UV aging test. An organoleptic examination also pointed that after the UV exposure, the color of PUR compositions altered from yellow to orange color which was most likely the cause the presence of the chemical substances of nutmeg which are responsive to the oxidation process, such as aliphatic polyesters or polyethers (Table 7). The structure of PUR samples with nutmeg filler was characterized by a smaller quantity of holes and cracks (Figure 8).

The obtained effect of the color analysis of PUR composite after UV aging are shown in Table 6. The most noticeable color change (determined by ΔE*) was visible for controlled PU_N_0 foam after the UV exposure. For PUR samples, the change in ΔE* before and after the UV exposure was diminished (Figure 9). The scientists say unequivocally concluded that the application of nutmeg filler may efficiently increase the anti-aging features of PUR samples due to the chemical structure of nutmeg seeds which engaged organic substances, such as polyphenols, quinones, or flavonoids. Moreover, it was concluded that the application of nutmeg filler may meaningly improve the stabilization of the PUR samples and may be added as an anti-aging substance in the production of PUR foams.

### 3.3. Polyurethane Foams Reinforced with Syzygium Aromaticum Filler

As another example of the use of organic fillers to reinforce the permanent polyurethane materials, *Syzygium aromaticum*, commonly known as cloves [9] is given. In the presented study, grounded cloves were used as cellulosic fillers for novel PUR composite foams. Soybean-oil-based PU composite foams were successfully reinforced with different concentrations (1, 2, and 5 wt.%) of clove filler. The reference foam was labeled as PU_0, whereas the PUR composite foams filled with 1, 2, and 5 wt.% of clove filler were labeled as PU_1, PU_2, and PU_5, respectively.

It was found that with increasing filler content, the morphology of PUR composites became less uniform with a higher number of damaged cells (Figure 10). It was also visible that the filler particles were not well built in the cell struts, but some particles were also visible in voids. The authors concluded that the incorporation of a high amount of clove filler may result in the formation of some aggregates, which in turn promotes the formation of broken cells.

The applied microbiological analyzed was carried out to estimate the action of clove filler on the antimicrobial characteristic of PUR samples (Table 8). A meaningful removal of selected bacteria—*E. coli* and *S.*
*aureus* was seen after 24 h of exposure. A sample containing 5 wt.% of clove filler had the best results. The reduction in the selected bacteria using 5 wt.% of clove filler was about 76 and 79%, which clearly indicates a strong fungicidal and microbicidal effect, protecting composites against bacteria and fungi.

In the next step, PUR foams were subjected to UV aging. The most visible change of total color changes (ΔE*) after the UV aging process was observed for PU-0 (Table 9). With the increasing content of clove filler, the difference in ΔE* was reduced. The high resistance of cloves against high temperature and UV radiation was attributed to the chemical composition of cloves, which included various extracts, such as eugenol, ether, or methanolic extracts with strong antioxidant activities.

An organoleptic test showed that after the UV radiation use, the color of PUR sample surface changed to orange meaningful even before the application of clove addition (Figure 11, Figure 12 and Figure 13). In all foams, the surface of PUR sample became less fragile after UV radiation than the appearance of the non-modified surface of PU-0. The area of PU-0 exposed to one week of degradation was more damaged and had a higher number of cracks and holes compared with PUR composites with the application of clove filler. The authors explained that the incorporation of clove filler may improve the strength of the bonds responsible for reducing the brittleness and improve the aging properties of PUR composites.

The mechanical efficiency of PUR foams was estimated as well. Compression strength (σ_10%_) of PUR samples was defined in a parallel and perpendicular direction in relation to the foam growth. The application of 1 and 2 wt.% of clove filler led to a meaning refinement of σ_10%_ which is not seen in the unmodified sample. The result of σ_10%_ (measured parallel) enhanced by ~18 and ~13% for composite PU-1 and PU-2. Another improvement of σ_10%_ is not observed for copositePU-5. For PU-0 the result of σ_10%_ changed by ~16%. Such observations were for flexural strength (σ_f_) and impact strength of PUR samples. The properties of samples were improved, but with increasing filler amount the mechanical feature decreased. Analyzing to PU-0, σ_f_ increased by ~11%, ~9%, and ~2% for composites PU-1, PU-2, and PU-5, but the impact strength was enhanced by ~18%, ~11%, and ~8%. A similar tendency was observed in the case of dynamic mechanical results (Figure 14). The incorporation of 1 and 2 wt.% of clove filler did not affect the value of the glass transition temperature (T_g_). The lower value of T_g_ was observed for sample PU-5. Based on this, it was concluded that the incorporation of clove filler in high concentrations, such as 5 wt.%, decreases the cross-linking density. On the other hand, the incorporation of 1 and 2 wt.% of clove filler improved the thermo-mechanical properties of PUR composite foams. According to the presented results, samples PU-1 and PU-2 exhibited a higher value of storage modulus (E’) compared with PU-0. When increasing clove filler content up to 5 wt.%, the value of E’ slightly decreased.

Nowadays, PUR antibacterial foams have been achieved in two ways. On the one hand, materials have been synthesized by applying an antibacterial, naturally occurring fillers or extracts. So far, several various natural modifiers have been investigated: nutmeg [10], curcumin [12], chitosan [28,29], cloves [9], lavender [13]. The use of the natural extracts and fillers received from simple raw sources such as plants, herbs, bio waste are very interesting from a scientific point of view because natural additions improve the environmentally friendly character of PUR foams. On the other hand, metal and a metal oxide, sulphide nanoparticles (NPs) with antimicrobial properties have been embedded into the structure or surface of the foam. It is worth giving an example such as Ag [60,61,62,63,64,65], Cu, Zn, and its oxides [66], CdS [67]. From the mentioned examples, Ag nanoparticles are most often used as antiseptic material in medical applications, as the presence of Ag poses a low danger of toxicity toward humans. Additionally, Ag has meaningful toxicity to many species of bacteria, fungi, and viruses. Zeolites are most recently used as natural or synthetic fillers, having hydrated alumina-silica structures of alkaline and alkaline-earth metals [68,69]. The chemical construction of natural zeolites lets for metallic ion loading or, due to their high characteristic surface, they are right for immobilizing metallic nanoparticles on zeolite area. Okuyama et al. [70] introduced a new method for obtaining antibacterial flexible polyurethane foam with Ag ion-charged synthetic zeolite. Czel et al. [71] presented instead new method for the synthesis of the open-cell soft polyurethane foam with silver nanoparticles on the surface of ultrafine grain natural zeolite particles as fillers. Presented research confirmed the polyurethane matrix with the nano-silver particles having a positive biocide effect. The antibacterial action of the PUR composites with natural zeolites containing Ag nanoparticles were examined against *Escherichia coli* (Gram-negative), and *Micrococcus luteus* (Gram-positive) in a 24 and 72 h period. The obtained scores indicate that the natural zeolite with Ag nanoparticles has an antibacterial effect, particularly against Gram-positive bacteria. All applied antimicrobial modifiers can stop the increase in bacteria, mold, and mildew in and on polyurethane foams. Restraining the multiplication of microbes in polyurethane foam, decrease microbial-related odors, provide esthetic appeal, and reduce degradation. Finally, this prolongs product life, commercial savings, and results in less material going to the garbage since articles do not need to be replaced as often. Table 10 summarizes different methods of polyurethane foams modification to obtain antibacterial properties of the material.

## 4. Conclusions

Polyurethanes are one of the common polymeric materials with great control over their physicochemical features based on chemical composition. The properties of polyurethanes enable them to obtain their various forms and to be applied in universally accessible products where are required conveniences, durability, warmth, and sound isolation.

Advisable research of scientists indicated that the chemical inclusion of modifying substances with assured antibacterial and antifungal properties into nonmodified polyurethane formulations enable obtain of antimicrobial products. The scientific description confirmed that the use of an antibacterial modifier can change the rheological behavior, cellular structure, and further mechanical properties of polyurethane materials.

Naturally occurring antimicrobials have gained attention among researchers and material manufacturer due to their safety, easily accessible and nontoxic status. Natural additives as extracts are easy to obtain from various plants. Fruit, vegetables, herbs, and spices have been found to be rich sources of aldehydes, ester terpenoids, phenolics, and sulfur-containing compounds. These natural occurring agents commonly found in roots, flowers, leaves, seeds and bulbs and in other parts of the plants. These substances are produced in defensive mechanism and are helpful for inactivation or inhibition of many microorganisms (bacteria, yeast and molds) and there have not harmful impacts on human health. These naturally occurring antimicrobial agents can be isolated from local sources using various advanced techniques. Due to potential adverse effect of some synthetic fungicides and antibacterial additives on environment and health there is a strong societal emphasis for use less synthetic antimicrobial additives in materials and the use of natural antimicrobial as alternatives. Despite the fact that natural additives have many advantages and can be successfully used as biocides, it must also be remembered that natural extracts may be less active compared with chemical equivalents, and have a shorter period of action. However, the presented research shows that this topic is very interesting because it concerns the use of organic substances in polymer materials [72,73,74,75,76,77,78,79,80,81,82]. The antimicrobial properties of the polyurethane were rated toward *E.coli* and *S. aureus* and others. The methods of polyurethane modification shown in this review may lead to obtaining materials with antibacterial and antifungal properties. Furthermore, this means ways for the formation of antibacterial and antifungal materials in different forms facilitate the future development of new safer, and more effective antibacterial polyurethane materials in the new applications in various fields. So far, there are not many reports on the use of natural antibacterial substances in foams, and the topic is constantly evolving because bacteria and fungi can infect walls, ceilings, and other places in commercial areas, causing unsightly conditions and even breathing problems for the building’s occupants. Frequently, everyday-use products are exposed to fungus and mold which cause ugly black and brown blots or greenish marks which can destroy the condition of materials if they are not protected with an antifungal modifier. Constantly endangering fungus and mold can cause products to fast decrease in product quality, stability, and product shelf life. The presented review confirmed that incorporation of natural modifiers with given antibacterial properties into conventional polyurethane and polyurethane foam formulations allows preparing of antimicrobial materials with effective alternatives to expensive and currently technologies [83,84,85,86,87,88,89,90,91,92,93,94,95,96,97,98,99,100,101,102].

Polymer materials have been widely used in various industries. The related need for microbiological protection forces the biostabilization of polymers and their composites. The scope of application of polymeric materials with microbiological protection, additionally enriched with antimicrobial properties, is very extensive and covers many industries. The most important are the medical industry, textile, packaging, water filtration systems, air conditioning. Protection of polymers against adhesion of microorganisms to the surface, as well as the colonization of products by them, are commonly used. The use of biocides for plastics prevents their discoloration, problems with maintaining microbiological cleanliness of the surface, slows down aging.

Summarizing our in-depth analysis of the natural antimicrobial substances used so far as additives to polyurethanes, we consider that the presented results prove that it can be potential strategy or candidate to be further applied extensively. The possibility of using such additives is confirmed not only by the fact of their effectiveness, but also the nature of their origin, the possibility of using, in many cases, bio-waste with antimicrobial properties. There are many natural antibiotics in nature; thus, further research should be carried out in this field because there are few such additives to polyurethanes described so far.

## Figures and Tables

**Figure 1 polymers-14-02533-f001:**
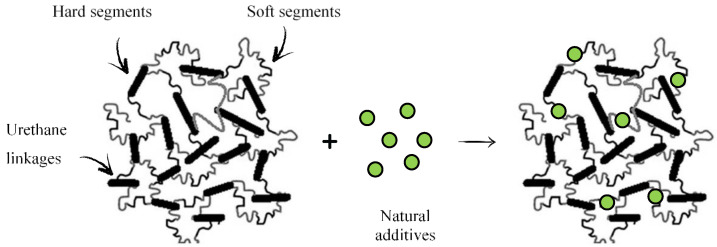
The impact of the natural additives’ addition on the crosslinking density of PUR composites. Adapted from Ref. [14].

**Figure 2 polymers-14-02533-f002:**
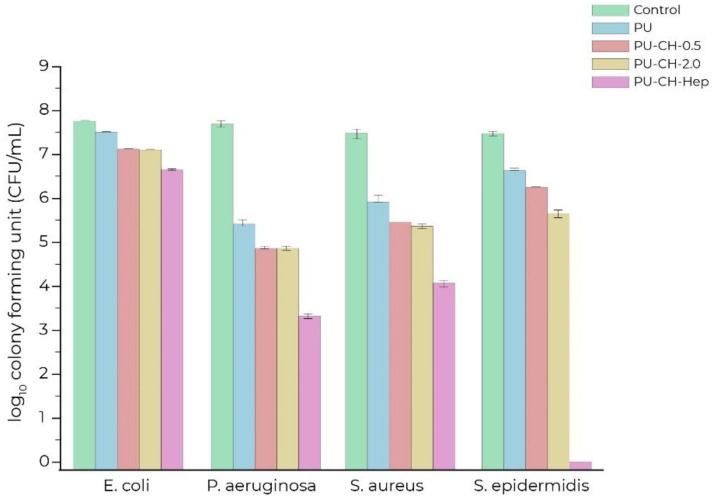
Antibacterial effects on control sample (Control), polyurethanes (PU), chitosan (CH), and heparin (Hep) immobilized on polyurethane films on survival of *E. coli, P. aeruginosa, S. aureus and S. epidermidis* bacteria. Adapted from Ref. [29].

**Figure 3 polymers-14-02533-f003:**
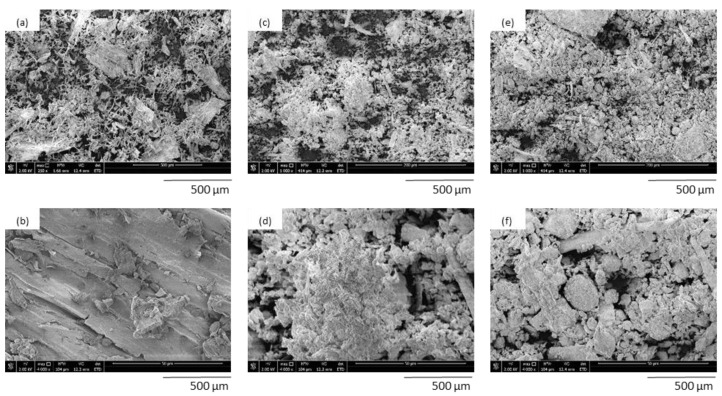
SEM images of lavender fillers: (**a**,**b**) non-functionalized lavender, (**c**,**d**) lavender functionalized with kaolinite, (**e**,**f**) lavender functionalized with hydroxyapatite. Adapted from Ref. [13].

**Figure 4 polymers-14-02533-f004:**
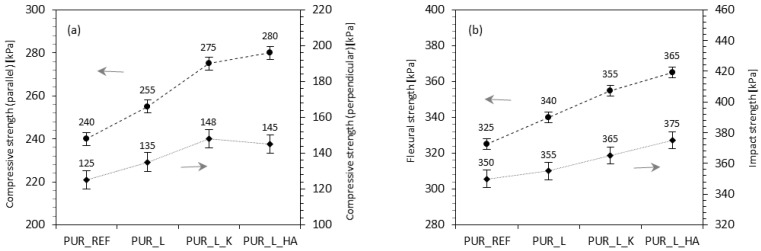
The mechanical performances of PUR foams: (**a**) compressive strength, (**b**) flexural strength, and impact strength. Adapted from Ref. [13].

**Figure 5 polymers-14-02533-f005:**
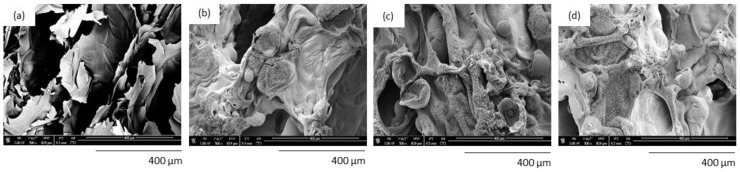
SEM images of char residue of (**a**) PUR_REF, (**b**) PUR_L, (**c**) PUR_L_K, and (**d**) PUR_L_HA (obtained after the cone calorimeter test). Adapted from Ref. [13].

**Figure 6 polymers-14-02533-f006:**
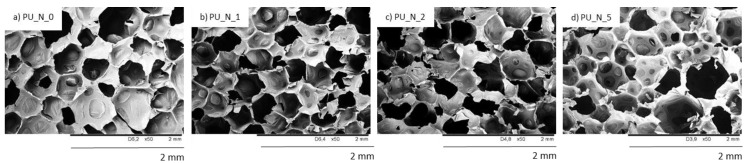
The cellular structure of (**a**) PU_N_0, (**b**) PU_N_1, (**c**) PU_N_2 and (**d**) PU_N_5 observed at a magnification of ×50. Adapted from Ref. [10].

**Figure 7 polymers-14-02533-f007:**
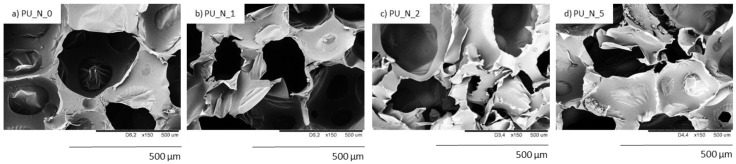
The cellular structure of (**a**) PU_N_0, (**b**) PU_N_1, (**c**) PU_N_2 and (**d**) PU_N_5 observed at a magnification of ×150. Adapted from Ref. [10].

**Figure 8 polymers-14-02533-f008:**
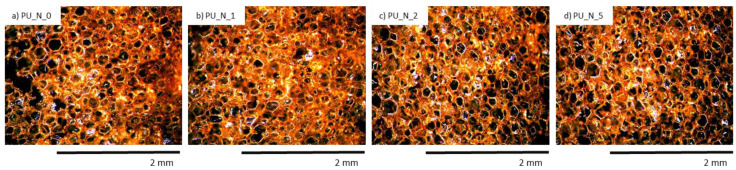
Morphology of PUR composite foams after 7 days of UV aging. Adapted from Ref. [10].

**Figure 9 polymers-14-02533-f009:**
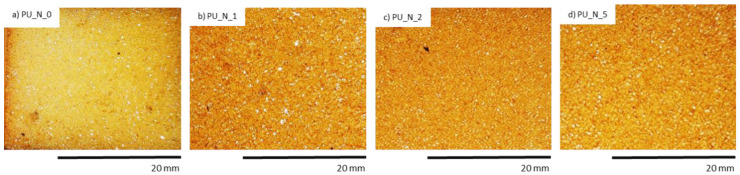
PUR composite foams after the 7 days of UV-aging. Adapted from Ref. [10].

**Figure 10 polymers-14-02533-f010:**
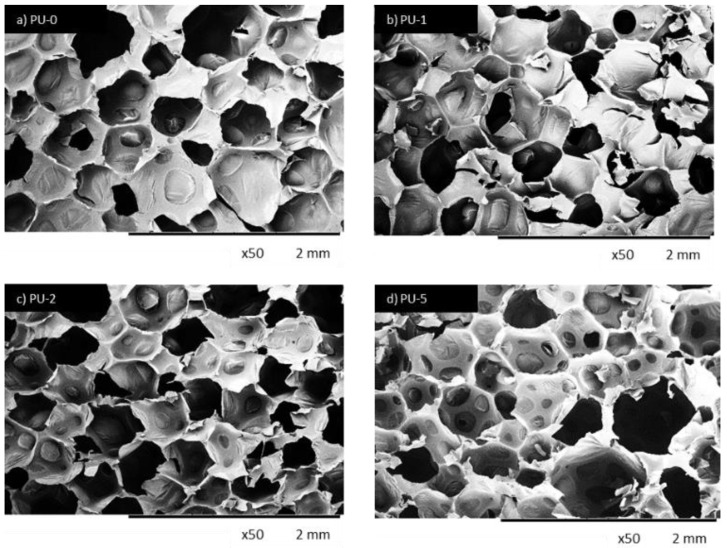
The cellular structure of (**a**) PU-0, (**b**) PU-1, (**c**) PU-2 and (**d**) PU-5 observed at a magnification of ×50. Adapted from Ref. [9].

**Figure 11 polymers-14-02533-f011:**
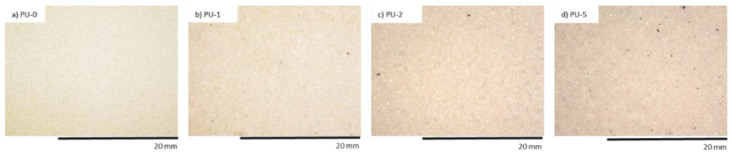
The external surface of PUR composite foams before UV aging. Adapted from Ref. [9].

**Figure 12 polymers-14-02533-f012:**
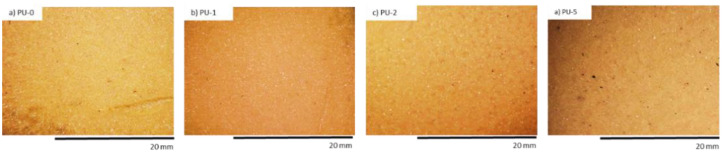
The external surface of PUR composite foams after 7 days of UV aging. Adapted from Ref. [9].

**Figure 13 polymers-14-02533-f013:**
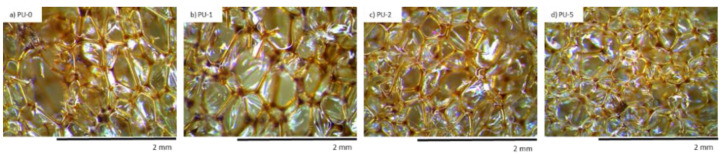
Morphology of PUR composite foams after 7 days of UV aging observed at a magnification of ×50. Adapted from Ref. [9].

**Figure 14 polymers-14-02533-f014:**
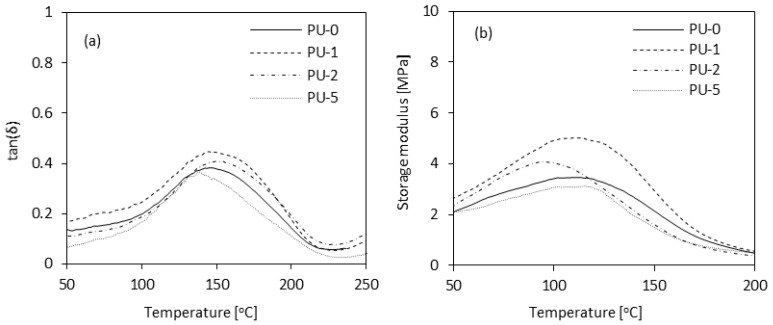
Dynamic mechanical analysis of PUR composite foams—(**a**) Tanδ and (**b**) storage modulus as a function of temperature. Adapted from Ref. [9].

**Table 1 polymers-14-02533-t001:** Inhibition areas (mm) of clove oil-coated nanofiber materials against *S. aureus* and *E. coli.* [30].

Selected Bacteria	Diameter of Area (mm) at a Different Quantity of Clove Oil			
	2 mg cm^−2^ Clove Oil Coated Polyurethane	5 mg cm^−^^2^ Clove Oil Coated Polyurethane	10 mg cm^−^^2^ Clove Oil Coated Polyurethane	Polyurethane
*S. aureus*	24	28	29	Not observed
*E. coli*	22	28	33	Not observed

**Table 2 polymers-14-02533-t002:** Methods of polyurethane modification to obtain antibacterial properties of the material.

Type of Additives	Concentration [%]	Bacterial Growth Inhibition				References
		*S. aureus*	*E. coli*	*P. aeruginosa*	*S. epidermidis*	
Cinnamaldehyde	0.5	−	−	not tested	not tested	[19]
Cinnamaldehyde	1	−	−	not tested	not tested	[19]
Cinnamaldehyde	2.5	+	+	not tested	not tested	[19]
Cinnamaldehyde	3.5	+	+	not tested	not tested	[19]
Cinnamaldehyde	5	+	+	not tested	not tested	[19]
Chitosan	0.5	+	−	+	+	[28]
Chitosan	2	+	−	+	+	[28]
Heparin	0.5	+	+	+	+	[28]
Thymol	4	−	−	not tested	not tested	[31]
Thymol	6	−	−	not tested	not tested	[31]
Thymol	8	+	+	not tested	not tested	[31]
Thymol	14	+	+	not tested	not tested	[31]
Salvia	3	+	+	+	not tested	[32]
Propolis	1	+	+	not tested	not tested	[35,38]
Propolis	2	+	+	not tested	not tested	[35,38]
Malva extract	15	+	+	not tested	not tested	[51,52]

+ fights bacteria; − not fights bacteria

**Table 3 polymers-14-02533-t003:** Antibacterial properties of PUR composites against selected bacteria and fungi. [13].

Sample	Bacteria		Fungi		
	*E. coli*	*S. aureus*	*B. subtilis*	*C. albicans*	*A. niger*
PUR_REF	−	−	−	−	−
PUR_L	+	+	+	−	−
PUR_L_K	+	+	+	−	−
PUR_L_HA	+	+	+	−	−

+ fights bacteria; − not fights bacteria.

**Table 4 polymers-14-02533-t004:** Flame retardant properties of PUR composites [13].

Sample	IT (s)	pHRR (kW m^−2^)	TSR (m^2^ m^−2^)	THR (MJ m^−2^)	COY/CO_2_Y (−)	LOI (%)
PUR_REF	4	263	1500	21.5	0.875	20.2
PUR_L	4	203	1400	21.1	0.844	20.5
PUR_L_K	6	144	1060	20.5	0.736	22.2
PUR_L_HA	6	130	1055	19.8	0.788	22.7

**Table 5 polymers-14-02533-t005:** Microbial properties of PU composite foams against Staphylococcus aureus and Escherichia coli. [10].

	Suspension of *S. aureus* (CFU/mL)		Suspension of *E. coli* (CFU/mL)	
	Initial	After 24 h of Exposure	Initial	After 24 h of Exposure
PU_N_0	76 × 10^6^	76 × 10^6^	76 × 10^6^	74 × 10^6^
PU_N_1	76 × 10^6^	38 × 10^6^	76 × 10^6^	40 × 10^6^
PU_N_2	76 × 10^6^	30 × 10^6^	76 × 10^6^	35 × 10^6^
PU_N_5	76 × 10^6^	15 × 10^6^	76 × 10^6^	9 × 10^6^

**Table 6 polymers-14-02533-t006:** Color analysis of PU foams before UV aging, where ΔE*—total color change, L*—degree of lightness, a*—red/green parameter, b*—yellow/blue parameter. [10].

	Colorimetric Parameters			
	L*	a*	b*	ΔE*
PU_N_0	10.4	22.4	−4.8	7.1
PU_N_1	23.2	42.5	−4.1	17.2
PU_N_2	38.5	45.4	−2.8	24.0
PU_N_5	42.5	43.5	−1.0	30.2

**Table 7 polymers-14-02533-t007:** Color analysis of PU foams after UV aging, where ΔE*—total color change, L*—degree of lightness, a*—red/green parameter, b*—yellow/blue parameter. [10].

	Colorimetric Parameters			
	L*	a*	b*	ΔE*
PU_N_0	19.5	23.4	−4.1	23.2
PU_N_1	29.2	78.1	−3.0	24.1
PU_N_2	49.9	78.3	−2.9	27.5
PU_N_5	58.5	79.3	0.5	29.5

**Table 8 polymers-14-02533-t008:** Microbial properties of PUR composite foams against *Staphylococcus aureus* and *Escherichia coli.* [9].

	*S. aureus*		*E. coli*	
	Initial Bacterial Suspension (CFU/mL)	Bacterial Suspension Measured after 24 h (CFU/mL)	Initial Bacterial Suspension (CFU/mL)	Bacterial Suspension Measured after 24 h (CFU/mL)
PU-0	74 × 10^6^	74 × 10^6^	74 × 10^6^	74 × 10^6^
PU-1	74 × 10^6^	48 × 10^6^	74 × 10^6^	42 × 10^6^
PU-2	74 × 10^6^	33 × 10^6^	74 × 10^6^	28 × 10^6^
PU-5	74 × 10^6^	18 × 10^6^	74 × 10^6^	16 × 10^6^

**Table 9 polymers-14-02533-t009:** Color analysis of PUR foams after UV aging, where ΔE*—total color change, L*—degree of lightness, a*—red/green parameter, b*—yellow/blue parameter. [9].

	Colorimetric Parameters			
	L*	a*	b*	ΔE*
PU-0	18.7	24.2	−4.2	22.5
PU-1	28.3	75.4	−3.1	23.2
PU-2	49.5	75.1	−2.5	28.6
PU-5	60.5	79.2	0.3	30.4

**Table 10 polymers-14-02533-t010:** Methods of polyurethane foams modification to obtain antibacterial properties of the material.

Type of Additives	Concentrations [%]	Bacterial Growth Inhibition				References
		*S. aureus*	*E. coli*	*P. aeruginosa*	*K. pneumoniae*	
Cinnamon extract	10	+	+	not tested	−	[54]
Lanender	2	+	+	not tested	not tested	[57,58]
Pine oil	10	+	+	not tested	not tested	[55]
Pine oil	15	+	+	not tested	not tested	[55]
Nutmeg	2	+	+	not tested	not tested	[10,59,60]
Nutmeg	5	+	+	not tested	not tested	[10,59,60]
Clove	2	+	+	not tested	not tested	[9]
Clove	5	+	+	not tested	not tested	[9]

+ fights bacteria; − does not fight bacteria.

## Data Availability

Not applicable.

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
