# Peer review of "Natural Additives Improving Polyurethane Antimicrobial Activity"

_polymers, 2022, doi:10.3390/polym14132533_

Round 1
Reviewer 1 Report
This review tries to summary and discuss the recent advances and understandings about enhancing the anti-microbial activities of polyurethane and polyurethane foams by adding various natural Additives to the PUR. In general, this manuscript is well written but I have some suggestions to may help to improve this manuscript.
Point 1:
Page 1, line 18. In general, antibiotics could be simply divided into antibacterial antibiotics or antifungal antibiotic. Thus, I suggest the authors rephrase your descriptions regarding “Antibiotics fight bacterial infections by killing bacteria (bactericidal effect) or slowing and stopping their growth (bacteriostatic effect), but they do not fight fungi and viruses, therefore compounds of natural origin can find a wide use as biocidal substances”. For example, “amphotericin B deoxycholate”, a polyene antibiotic, was the first antimycotic agent introduced in 1958 to treat systemic mycoses.
Point 2: the authors have described quite a few methods to could raise anti-microbial activities of PUR at page 3 to 7. I strongly recommend the authors use a table to summary previous findings to let the readers understand those advances easily. And, this table should include the important findings about the dosage, concentrations, mechanisms, approaches and references of employed natural additives in this field.
Point 3: the authors have described quite a few methods to could raise antiseptic properties of PUR foam at page 8 to 9. I strongly recommend the authors use a table to summary previous important findings to let the readers understand those advances easily. And, this table should include the important findings about the dosage, concentrations, mechanisms, approaches and references of employed natural additives to could elevate the anti-microbial activities of PUR foam.
Point 4: as for the conclusion section (page 17), I recommend the authors discuss more about their view points about which kinds or types of naturally occurring antimicrobials or approaches could be the most potential strategy or candidate to be further applied extensively. Only to summary previous findings in brief again in this section may not be enough.
Reviewer 2 Report
The manuscript titled "Natural Additives Improving Polyurethane Antimicrobial Activity” (polymers-1740814) provides an overview of the additives that could be incorporated into polyurethanes to turn them into possible antimicrobial materials. Overall, in my opinion, the manuscript lacks organization, clarity in presentation, and extensive English revision. For those reasons, I would reconsider its publication in "Polymers" journal after major revisions. My suggestions for the improvement of the manuscript are mentioned below.
Major revisions:
1. Overall, the writing across the entire manuscript lacks clarity and would strongly benefit from a revision. I would recommend having a 3rd party (English native) read and edit the manuscript.
2. Scale bars are missing or are difficult to see in many figures (eg. Figure 3, figure 5, figure 8, figure 9). Please add them.
3. Chapters 2 and 3 have the following titles “ Antimicrobial polyurethane” and "antimicrobial polyurethane foams". In my opinion those are misleading titles since polyurethanes by themselves do not present antimicrobial activity. My recommendation is to alter both titles.
4. Chapter 2 and 3 are quite extensive and would benefit from subdivisions that for example, could divide different types of fillers and additives that are incorporated into the PURs. By doing this, the clarity and quality of presentation of the manuscript would be improved and the entire document would be easier to understand and more appealing to readers.
Minor revisions:
- Reference style is inconsistent. Please revise it.
Round 2
Reviewer 1 Report
The revised version of the manuscript has been considerately improved.
Reviewer 2 Report
The authors have addressed all my comments. In my opinion, the manuscript should be published in its present form.